# Exploration of Metabolic Biomarkers Linking Red Meat Consumption to Ischemic Heart Disease Mortality in the UK Biobank

**DOI:** 10.3390/nu15081865

**Published:** 2023-04-13

**Authors:** Bohan Fan, Xin Huang, Jie V. Zhao

**Affiliations:** School of Public Health, Li Ka Shing Faculty of Medicine, The University of Hong Kong, Pokfulam Road, Hong Kong SAR, China

**Keywords:** red meat, metabolic factors, cardiovascular health

## Abstract

Growing evidence suggests that red meat consumption is a risk factor for cardiovascular health, with potential sex disparity. The metabolic mechanisms have not been fully understood. Using the UK Biobank, first we examined the associations of unprocessed red meat and processed meat with ischemic heart disease (IHD) mortality overall and by sex using logistic regression. Then, we examined the overall and sex-specific associations of red meat consumption with metabolites using multivariable regression, as well as the associations of selected metabolites with IHD mortality using logistic regression. We further selected metabolic biomarkers that are linked to both red meat consumption and IHD, with concordant directions. Unprocessed red meat and processed meat consumption was associated with higher IHD mortality overall and in men. Thirteen metabolites were associated with both unprocessed red meat and IHD mortality overall and showed a consistent direction, including triglycerides in different lipoproteins, phospholipids in very small very-low-density lipoprotein (VLDL), docosahexaenoic acid, tyrosine, creatinine, glucose, and glycoprotein acetyls. Ten metabolites related to triglycerides and VLDL were positively associated with both unprocessed red meat consumption and IHD mortality in men, but not in women. Processed meat consumption showed similar results with unprocessed red meat. Triglycerides in lipoproteins, fatty acids, and some nonlipid metabolites may play a role linking meat consumption to IHD. Triglycerides and VLDL-related lipid metabolism may contribute to the sex-specific associations. Sex differences should be considered in dietary recommendations.

## 1. Introduction

Ischemic heart disease (IHD), the leading subtype of cardiovascular disease, is a major public health challenge, resulting in over 9 million deaths globally in 2019 [1]. At the global level, the morbidity and mortality of IHD in men are substantially higher than in women and are still rising rapidly [1]. Dietary factors are important in IHD prevention, as dietary habits have been shown to influence multiple biological pathways, including lipoprotein and cholesterol metabolisms, blood pressure, and overall cardiovascular health [2,3].

Concern is growing regarding the consumption of red meat. A meta-analysis concerning 1,427,989 people reported that a higher consumption of unprocessed and processed red meat was associated with a 9% and 18% increased risk of coronary heart disease (CHD), respectively [4]. Based on the existing evidence, mainstream dietary recommendations suggest minimizing the consumption of red meat and saturated fat for optimal health [5,6,7]. Although the UK population has cut its meat consumption by 17% over the past decade, a significant proportion of the population (34%) still consumes more red meat than recommended [8]. Red meat has large amounts of saturated fatty acids (SFAs), which is generally considered as a risk factor for dyslipidemia [9]. Nevertheless, more critical evidence is needed to explore the underlying mechanisms, especially as the role of SFAs in red meat in CHD risk remains inconsistent at present [9,10].

Changes in metabolic profiles are closely linked to IHD mortality [11]. For example, previous studies have highlighted the critical role of lipid and fatty acid components in atherosclerotic plaque formation and subsequent IHD development, including cholesterol, phosphatidic acids, phosphatidylcholine, etc. [12,13,14]. Recent advancements of nuclear magnetic resonance (NMR) metabolomics allow the investigation of diet-related metabolic alterations and their associations with disease risk, which can help to elucidate the metabolic mechanisms underlying the relationship between red meat and IHD risk [15]. The systematic analysis of small-molecule metabolites in NMR metabolomics provides a more comprehensive way to assess the mechanistic links for the associations [16]. A recent nested case-control study based on data of NMR metabolomics of a Chinese population reported that 29 metabolites showed a directionally consistent association with red meat consumption and cardiovascular disease (CVD) risk, revealing the role of lipid-related metabolites in the detrimental effects of red meat on CVD risk [17]. Dietary patterns in China remarkably differ from those of European countries, and the per capita red meat consumption in European countries far exceeds that of China [18]. There is a lack of metabolomic evidence from European populations. In addition, men have been found to have more red meat consumption compared with women across Europe [19], and men may have a higher risk of IHD associated with red meat consumption compared with women in East Asians [20,21] but unclear in Europeans. It is necessary to examine the sex-specific role of red meat in IHD risk and to identify the metabolic mechanism behind the sex difference.

Taken together, the current study aimed to identify metabolic markers associated with both red meat consumption and with IHD mortality based on the NMR metabolomic profiling data from the UK Biobank, which covers a wide range of metabolic biomarkers, including lipoprotein lipids, fatty acids, and multiple low-molecular-weight metabolites. Considering the sex difference, we also tested the sex-specific associations of red meat consumption with IHD mortality and with metabolites, and the sex-specific associations of metabolites with IHD mortality, to identify which metabolic markers may explain the sex difference for the associations of red meat with IHD mortality.

## 2. Methods

### 2.1. Study Design

First, we tested the associations of unprocessed red meat and processed meat with IHD mortality overall, and then in men and women separately. Consistent with previous studies [20,21], we found red meat consumption was positively related to IHD overall and in men, but not in women (OR 1.01, 95% CI 0.95, 1.08) (Appendix A). Then, we examined the relationship of unprocessed red meat with metabolites and the associations of metabolites with IHD risk overall. We selected metabolic biomarkers that are both linked to red meat consumption and associated with IHD, with concordant directions. In sex-specific analysis, we identified metabolites, which are specifically associated with IHD in men, and metabolites which are linked to red meat consumption only in men but not in women. Likewise, we repeated the analysis with processed meat consumption to identify metabolic biomarkers associated with processed meat and with IHD, overall and by sex. The details of metabolite selection were shown in statistical analysis.

### 2.2. Study Population

The UK Biobank is a large population-based cohort study that recruited ~500,000 participants aged 40–69 years from 2006–2010 across the UK [22], with an average follow-up time of 14 years. At recruitment, participants completed touchscreen questionnaires, verbal interviews, physical measurements, and biosample collections at baseline assessment centers. The touchscreen questionnaires included information on sociodemographic factors, lifestyle, personal health status, and family medical history. Electronic informed consent was obtained from all participants.

### 2.3. Assessment of Meat Consumption

The baseline touchscreen questionnaire asked about meat intake including processed meat (such as sausages, chicken nuggets, kebabs, bacon, burgers, ham, and meat pies), unprocessed beef, unprocessed lamb/mutton, and unprocessed pork. The food intake consisted of 6 categories, and we converted them into consumption frequencies: never eaten, eaten < 1 time per week, 1 time per week, 2–4 times per week, 5–6 times per week, and ≥1 time daily, and assigned them values of 0, 0.5, 1, 3, 5.5, and 7, respectively. We totalled unprocessed beef, lamb/mutton, and pork frequencies of consumption into unprocessed red meat consumption.

### 2.4. Metabolomic Profiling

High-throughput targeted NMR spectrometry was used to conducted metabolomic profiling on a subset of ~120,000 randomly selected plasma samples. Details of the methodology are described here (https://nightingalehealth.com/ (accessed on 20 July 2022)). A total of 249 metabolites were measured between June 2019 and April 2020. They were grouped based on metabolic pathways, including lipoprotein lipids in 14 subclasses, amino acids, fatty acids, inflammation biomarkers, fluid balance markers, glycolysis metabolites, and ketone bodies. We included 167 metabolites that were directly measured and quantified in mmol/l, and each metabolite was standardized prior to analysis.

### 2.5. IHD Mortality

IHD cases in the UK Biobank were identified by linking to data from death registry records, which were defined according to the International Classification of Diseases (ICD) as I20-I25.

### 2.6. Confounders

Sociodemographic and lifestyle-related information was extracted from the UK Biobank baseline assessment. This included age at recruitment, sex, ethnic background, education, alcohol drinking status, smoking status, physical activity, Townsend deprivation index (TDI), medications use, baseline disease, and body mass index (BMI). Educational levels were categorized as with or without college/university degree. Self-reported smoking status and alcohol drinking status were categorized into current users, previous users, and nonusers. Physical activity measurements were categorized into low, moderate, and high levels. TDI was calculated based on the percentages of nonhome owners, unemployment, households without car ownership, and overcrowded households [23]. Medications use (Yes/No) were obtained from use of medications for diabetes, blood pressure, and cholesterol or the use of exogenous hormones as well as of other prescription medications. The baseline disease status was accounted for based on whether individuals with vascular/heart problems or diabetes were diagnosed by a doctor. BMI was obtained by dividing weight by standing height^2^ (in kg/m^2^) based on measurements from the baseline survey.

### 2.7. Statistical Analyses

We first tested the association of unprocessed red meat consumption with IHD mortality overall and by sex using logistic regression. We then investigated whether biomarkers could contribute to the associations of red meat consumption with IHD mortality, overall and sex specifically. Multivariable linear regression was used to assess the association of unprocessed red meat consumption with metabolites, and logistic regression was used to assess the association of metabolites with IHD mortality. The models adjusted for age, sex, ethnicity, education, alcohol status, smoking status, physical activity, TDI, medications use, and baseline disease status (model 1). Similarly, we performed the same analyses with processed meat consumption.

For overall association, we selected biomarkers, which were associated with both unprocessed red meat consumption and with IHD mortality that show consistent directions of effects (i.e., both positive or both negative in direction of associations of metabolites with red meat consumption and with IHD). For sex-specific association, we identified metabolites, which were associated with both unprocessed red meat consumption and with IHD mortality in men but were either unrelated to unprocessed red meat consumption in women or unrelated to IHD in women. These biomarkers must also have consistent directions of associations with IHD risk and with red meat consumption. In sensitivity analyses, we further explored associations of metabolites with unprocessed red meat and processed meat consumption and additionally adjusted for BMI (model 2). To control the false discovery rate in multiple testing, we utilized the Benjamini–Hochberg method [24]. All statistical analyses were performed using R (Version 4.0.1).

## 3. Results

Baseline sociodemographic and lifestyle characteristics of the UK Biobank study participants are summarized in Table 1. The mean (SD) age is 55.4 (8.09) years, and 52.1% are women. Overall, 74,713 participants were included in the analysis with unprocessed red meat, and 633 IHD mortalities were identified. Among 35,668 men, 515 died of IHD; among 38,763 women, 118 died of IHD. A similar pattern was observed in the analysis with processed meat.

A flow diagram of study participants included for analyses is presented in Appendix A. The distribution of 167 metabolites is shown in Appendix A.

Among all metabolites, 161 were associated with unprocessed red meat consumption, 157 with processed meat consumption (Appendix A), and 22 with IHD risk (Appendix A). Thirteen metabolites were associated with both unprocessed red meat and IHD risk overall (Table 2a). Specifically, we found overall unprocessed red meat consumption associated with higher triglycerides in different lipoproteins (very small very-low-density lipoprotein (VLDL), intermediate-density lipoprotein (IDL), large low-density lipoprotein (LDL), LDL, medium LDL, small LDL, very large high-density lipoprotein (HDL)), phospholipids in very small VLDL, creatinine, glucose, glycoprotein acetyls, and tyrosine that were linked to an increased IHD risk. In contrast, negative associations were observed for docosahexaenoic acid (DHA); that is, unprocessed red meat is linked to lower DHA levels, and DHA is associated with a lower IHD risk. Compared with unprocessed red meat, we obtained similar results for processed meat. We also found that processed meat consumption is associated with lower albumin and that albumin is related to a lower IHD risk (Table 2b).

Using a similar approach, we found 10 metabolites that were positively associated with both unprocessed red meat and IHD in men (Table 3a), but these metabolites were neither corresponding to red meat consumption nor related to IHD in women (Appendix A). These include concentrations of very small VLDL particles, phospholipids in very small VLDL, total lipids in very small VLDL, as well as triglycerides in IDL, large LDL, LDL, medium LDL, small LDL, very large HDL, and very small VLDL. The results are consistent for processed meat consumption (Table 3b and Appendix A). In sensitivity analyses, most associations with processed meat consumption were similar to those with unprocessed red meat consumption. Model 2 additionally adjusts for BMI and moves estimates toward null (Appendix A).

## 4. Discussion

Using data from the UK Biobank on dietary frequency, NMR metabolomics, and IHD death records, this study found that unprocessed and processed red meat consumption was associated with higher IHD mortality overall and in men. Thirteen metabolic biomarkers showed a directionally consistent and significant association with unprocessed red meat consumption and IHD mortality, including triglycerides in different lipoproteins, phospholipids in very small VLDL, DHA, tyrosine, creatinine, glucose, and glycoprotein acetyls. In addition, metabolic biomarkers related to the metabolism of triglycerides and VLDL may contribute to the sex difference as they were simultaneously related to both unprocessed red meat consumption and IHD mortality in men but not in women. We also obtained similar results for processed meat. Our work promotes the understanding of the metabolite profiles, which may explain the detrimental effect of red meat consumption on IHD risk in European populations and the higher risk in men.

We focused on plasma metabolic biomarkers linking red meat consumption to IHD mortality. Although there are compositional differences between unprocessed red meat and processed meat, our results suggest that they may influence the development of IHD through quite similar metabolic pathways. We found that triglycerides in seven lipoprotein subclasses (very small VLDL, IDL, large LDL, LDL, medium LDL, small LDL, and very large HDL) were positively associated with both red meat consumption and IHD mortality. Our findings are consistent with previous intervention studies that continuous consumption of red meat, such as pork and beef, affected plasma triglyceride levels [25,26]. Triglycerides are major components of triglyceride-rich lipoproteins (TRLs). TRLs can easily be deposited on the arterial walls and damage the endothelium, leading to atherosclerosis or the thickening of the arteries (arteriosclerosis), which is a well-known risk factor for cardiovascular diseases [27,28]. Chylomicron and VLDL particles are the main TRLs in all types of lipoproteins [29]. Triglycerides in smaller-sized lipoprotein particles (i.e., LDL, IDL, and HDL) can easily enter the arterial intima and become trapped and are associated with an atherogenic response [28,30,31,32]. The accumulation of triglycerides in different-sized lipoproteins induced by red meat consumption could biologically explain the increased risk of IHD. Our findings are consistent with the previous metabolomic analysis based on Chinese populations, in which red meat consumption has an effect on the myocardial infarction risk by influencing triglyceride and cholesterol transport [17]. Our results also show that red meat consumption was also relevant to elevated levels of other lipid risk factors for IHD, including apolipoproteins VLDL, IDL, LDL and their subfractions (including cholesterol, free cholesterol, cholesteryl esters, and phospholipids), and fatty acids.

In our study, DHA was negatively associated with both red meat consumption and IHD mortality. DHA is considered to be cardioprotective [33]. However, studies from China showed a positive association between total red meat consumption and DHA [17]. This may be related to the type of red meat, as our results show that compared to unprocessed red meat, processed meat had a stronger association with decreased DHA both before and after adjustment for BMI. Food processing may be contributed to alterations in the fatty acid composition [34], and a cross-sectional study also found that DHA levels were significantly lower in dietary patterns dominated by processed red meat [35].

In addition, we found that red meat was positively associated with tyrosine, creatinine, glucose, and glycoprotein acetyls, and these metabolites showed the same direction with IHD mortality. These findings suggest that red meat consumption may also link to a higher IHD risk through other nonlipid-related metabolisms. For example, red meat, as an important dietary source of amino acids, upregulates the metabolism of aromatic and branched-chain amino acids, which are associated with atherosclerotic plaques and coronary artery lesions [36,37,38]. Red meat may also mediate inflammation, glucose metabolism, and renal metabolism, and previous metabolomic studies have demonstrated the association of these metabolites with cardiovascular risks [38,39,40,41].

Interestingly, we conducted a sex-specific analysis and found that unprocessed red meat and processed meat consumption had a significant association with an increased risk of IHD mortality in men, but not in women. This was consistent with previous cohort studies showing that red meat intake is associated with a higher incidence of hypercholesterolemia and hyperlipidemia only in men [42]. Notably, our analysis of metabolites revealed important sex differences in the association of triglycerides of lipoproteins, very small VLDL, and its subfractions with red meat consumption and IHD mortality, which may be the endogenous contributor to sex differences in the effects of red meat. Previous studies have shown that there are important sex differences in lipid and lipoprotein metabolism possibly driven by the effects of sex hormones [43]. Women have been found to have an improved clearance of meal-related triglycerides due to liver estrogen signaling compared to men [43,44]. Estrogen may also enhance the cholesterol efflux by promoting the reverse cholesterol transport step in the liver, thus contributing to sex differences in atherosclerosis [43,45,46]. These results are of great public health importance, as the average meat intake of European men (84–218 g/day) far exceeds that of women (64–163 g/day), and the UK Dietary Guidelines recommend an intake of no more than 70 g of red or processed meat per day [19]. Therefore, considering the sex difference in metabolism, men should restrict red meat consumption and should follow a low-triglyceride diet and choose lean meat whenever possible.

The strengths of this study include the large sample size, involving nearly 80,000 participants from the UK biobank, considering both unprocessed red meat and processed meat consumption, and the consideration of sex difference in the associations, adding additional evidence to the mechanistic exploration of the overall and sex-specific associations of red meat consumption, metabolites, and IHD mortality. Our study also has some limitations. First, the observational study was unable to conclude a causal relationship because of residual confounding. However, this study provides useful information for future studies when long-term randomized controlled trials are unavailable. Second, red meat consumption was only assessed by the Food Frequency Questionnaire, and recall bias might exist. However, there was considerable consistency in the responses to the dietary questions about red meat at baseline and the repeat visit 4 years later [47]. Third, although the established targeted NMR metabolomic platform we used quantified a wide range of metabolic biomarkers and had the advantage of high specificity [48], it tests targeted metabolites; measurements of some metabolites, such as bioactive peptides, were not included. Last, our study was based on data of the UK population. Caution should be applied when generalizing our findings to other populations as even in the European region, there are differences in the amount and type of red meat consumption in different countries [19]. Considering that red meat consumption is quite large in most of Europe and that Europe is the second-largest meat consumer in the world [19,49], further studies to identify metabolites associated with red meat consumption and IHD in different countries are warranted to provide evidence for dietary guidance in European populations.

## 5. Conclusions

In summary, this study provided strong evidence to elucidate the metabolic mechanisms underlying the association of unprocessed red meat and processed meat consumption with IHD mortality in the UK population. We found that higher red meat consumption was associated with higher IHD mortality overall and in men. Triglycerides in lipoproteins, fatty acids, and some other nonlipid metabolites may play key roles in these associations. Triglycerides and VLDL-related lipid metabolism may be the endogenous contributor to sex differences in the role of red meat consumption in IHD. Our work supports the recommendation that both unprocessed red meat and processed meat consumption needs to be restricted in the UK and sex differences should be considered in the development of relevant dietary guidelines.

## Figures and Tables

**Table 1 nutrients-15-01865-t001:** Baseline characteristics of study participants.

Characteristics	Mean (SD) or %
**Mean processed meat consumption (SD), times/week**	1.47 (1.39)
**Mean unprocessed red meat consumption (SD), times/week**	2.10 (1.42)
**Mean age (SD), year**	55.4 (8.09)
**Women, %**	52.1
**Ethnic background, %**	
White	95.1
Asian or Asian British	1.8
Black or Black British	1.4
Chinese	0.3
Mixed	0.6
Other/unknown	0.8
**Townsend deprivation index (SD) ***	−1.57 (2.95)
**With college or university degree, %**	41.3
**Smoking status, %**	
Never	56.5
Previous	34.0
Current	9.5
**Alcohol status, %**	
Never	3.5
Previous	3.0
Current	93.5
**Physical activity, %**	
Low	18.9
Moderate	41.4
High	39.7
**Mean BMI (SD), kg/m^2^**	27.2 (4.67)
**Use of medications for cholesterol, blood pressure, and diabetes or use of exogenous hormones, %**	
No	71.1
Yes	28.9
**Use of other prescription medications, %**	
No	57.0
Yes	43.0
**Baseline disease (i.e., vascular/heart problems or diabetes diagnosed by doctor), %**	
No	71.9
Yes	28.1

* Townsend deprivation index, higher scores represent higher levels of deprivation. Sample size for processed meat = 74,713 and for unprocessed red meat = 74,431. Baseline characteristics of study participants by unprocessed red meat and by processed meat are similar, so, only data of unprocessed red meat are presented.

**Table 2 nutrients-15-01865-t002:** Overall association of metabolic biomarkers with red meat and with IHD.

Metabolites	Beta,95% CI_Meat	p_Meat	OR,95% CI_IHD	p_IHD
(a) Unprocessed red meat consumption				
Creatinine	0.027 (0.023, 0.031)	6.50 × 10^−34^	1.09 (1.05, 1.13)	9.82 × 10^−7^
Docosahexaenoic Acid	−0.005 (−0.01, −0.001)	2.37 × 10^−2^	0.88 (0.81, 0.96)	4.11 × 10^−3^
Glucose	0.011 (0.006, 0.015)	4.95 × 10^−6^	1.2 (1.16, 1.25)	5.26 × 10^−21^
Glycoprotein Acetyls	0.021 (0.016, 0.026)	1.55 × 10^−18^	1.24 (1.15, 1.33)	1.43 × 10^−8^
Phospholipids in Very Small VLDL	0.026 (0.021, 0.031)	2.97 × 10^−27^	1.14 (1.05, 1.23)	1.31 × 10^−3^
Triglycerides in IDL	0.028 (0.023, 0.032)	2.11 × 10^−29^	1.17 (1.09, 1.26)	1.04 × 10^−5^
Triglycerides in Large LDL	0.028 (0.023, 0.033)	2.55 × 10^−30^	1.18 (1.09, 1.26)	5.44 × 10^−6^
Triglycerides in LDL	0.027 (0.022, 0.032)	2.33 × 10^−28^	1.16 (1.08, 1.24)	2.25 × 10^−5^
Triglycerides in Medium LDL	0.025 (0.02, 0.029)	4.26 × 10^−24^	1.15 (1.07, 1.23)	6.51 × 10^−5^
Triglycerides in Small LDL	0.024 (0.019, 0.029)	7.91 × 10^−23^	1.11 (1.03, 1.19)	3.06 × 10^−3^
Triglycerides in Very Large HDL	0.018 (0.013, 0.023)	3.35 × 10^−13^	1.11 (1.04, 1.2)	2.30 × 10^−3^
Triglycerides in Very Small VLDL	0.025 (0.02, 0.03)	4.70 × 10^−25^	1.16 (1.08, 1.24)	5.45 × 10^−5^
Tyrosine	0.019 (0.014, 0.024)	8.76 × 10^−15^	1.13 (1.05, 1.21)	1.03 × 10^−3^
(b) Processed meat consumption				
Albumin	−0.018 (−0.023, −0.013)	4.18 × 10^−12^	0.82 (0.76, 0.89)	1.75 × 10^−7^
Creatinine	0.032 (0.027, 0.036)	8.19 × 10^−41^	1.09 (1.05, 1.13)	9.82 × 10^−7^
Docosahexaenoic Acid	−0.043 (−0.048, −0.038)	6.87 × 10^−66^	0.88 (0.81, 0.96)	4.11 × 10^−3^
Glucose	0.014 (0.009, 0.019)	1.16 × 10^−8^	1.2 (1.16, 1.25)	5.26 × 10^−21^
Glycoprotein Acetyls	0.037 (0.032, 0.042)	4.60 × 10^−49^	1.24 (1.15, 1.33)	1.43 × 10^−8^
Phospholipids in Very Small VLDL	0.02 (0.015, 0.025)	9.93 × 10^−15^	1.14 (1.05, 1.23)	1.31 × 10^−3^
Triglycerides in IDL	0.026 (0.021, 0.031)	6.71 × 10^−24^	1.17 (1.09, 1.26)	1.04 × 10^−5^
Triglycerides in Large LDL	0.028 (0.023, 0.033)	8.45 × 10^−27^	1.18 (1.09, 1.26)	5.44 × 10^−6^
Triglycerides in LDL	0.03 (0.024, 0.035)	2.32 × 10^−30^	1.16 (1.08, 1.24)	2.25 × 10^−5^
Triglycerides in Medium LDL	0.03 (0.025, 0.035)	7.62 × 10^−32^	1.15 (1.07, 1.23)	6.51 × 10^−5^
Triglycerides in Small LDL	0.034 (0.029, 0.039)	2.62 × 10^−39^	1.11 (1.03, 1.19)	3.06 × 10^−3^
Triglycerides in Very Large HDL	0.022 (0.017, 0.027)	4.78 × 10^−17^	1.11 (1.04, 1.2)	2.30 × 10^−3^
Triglycerides in Very Small VLDL	0.027 (0.022, 0.032)	2.20 × 10^−25^	1.16 (1.08, 1.24)	5.45 × 10^−5^
Tyrosine	0.013 (0.008, 0.018)	5.72 × 10^−7^	1.13 (1.05, 1.21)	1.03 × 10^−3^

Beta: per 1 SD unit increase in metabolites.

**Table 3 nutrients-15-01865-t003:** Association of 10 metabolic biomarkers with red meat and with ischemic heart disease in men.

Metabolites	Beta, 95% CI_Meat	p_Meat	OR, 95% CI_IHD	p_IHD
(a) Unprocessed red meat consumption				
Concentration of Very Small VLDL Particles	0.02 (0.02, 0.03)	1.62 × 10^−15^	1.14 (1.04, 1.25)	5.17 × 10^−3^
Phospholipids in Very Small VLDL	0.02 (0.02, 0.03)	5.93 × 10^−15^	1.17 (1.07, 1.28)	4.86 × 10^−4^
Total Lipids in Very Small VLDL	0.03 (0.02, 0.03)	3.19 × 10^−16^	1.15 (1.04, 1.26)	3.97 × 10^−3^
Triglycerides in IDL	0.03 (0.03, 0.04)	1.27 × 10^−20^	1.19 (1.1, 1.29)	1.01 × 10^−5^
Triglycerides in Large LDL	0.03 (0.03, 0.04)	5.27 × 10^−22^	1.2 (1.11, 1.29)	5.09 × 10^−6^
Triglycerides in LDL	0.03 (0.03, 0.04)	1.41 × 10^−20^	1.18 (1.09, 1.27)	2.32 × 10^−5^
Triglycerides in Medium LDL	0.03 (0.02, 0.04)	4.34 × 10^−18^	1.16 (1.08, 1.25)	7.15 × 10^−5^
Triglycerides in Small LDL	0.03 (0.02, 0.04)	2.85 × 10^−16^	1.12 (1.04, 1.21)	3.43 × 10^−3^
Triglycerides in Very Large HDL	0.02 (0.02, 0.03)	1.94 × 10^−11^	1.14 (1.05, 1.23)	9.14 × 10^−4^
Triglycerides in Very Small VLDL	0.03 (0.02, 0.04)	8.18 × 10^−17^	1.17 (1.08, 1.27)	1.08 × 10^−4^
(b) Processed meat consumption				
Concentration of Very Small VLDL Particles	0.01 (0.01, 0.02)	1.49 × 10^−6^	1.14 (1.04, 1.25)	5.17 × 10^−3^
Phospholipids in Very Small VLDL	0.02 (0.01, 0.02)	1.17 × 10^−6^	1.17 (1.07, 1.28)	4.86 × 10^−4^
Total Lipids in Very Small VLDL	0.02 (0.01, 0.02)	4.77 × 10^−7^	1.15 (1.04, 1.26)	3.97 × 10^−3^
Triglycerides in IDL	0.02 (0.02, 0.03)	2.04 × 10^−13^	1.19 (1.1, 1.29)	1.01 × 10^−5^
Triglycerides in Large LDL	0.03 (0.02, 0.03)	7.13 × 10^−16^	1.2 (1.11, 1.29)	5.09 × 10^−6^
Triglycerides in LDL	0.03 (0.02, 0.04)	2.86 × 10^−17^	1.18 (1.09, 1.27)	2.32 × 10^−5^
Triglycerides in Medium LDL	0.03 (0.02, 0.04)	1.13 × 10^−17^	1.16 (1.08, 1.25)	7.15 × 10^−5^
Triglycerides in Small LDL	0.03 (0.03, 0.04)	3.33 × 10^−20^	1.12 (1.04, 1.21)	3.43 × 10^−3^
Triglycerides in Very Large HDL	0.02 (0.02, 0.03)	7.61 × 10^−12^	1.14 (1.05, 1.23)	9.14 × 10^−4^
Triglycerides in Very Small VLDL	0.02 (0.02, 0.03)	2.04 × 10^−12^	1.17 (1.08, 1.27)	1.08 × 10^−4^

Beta: per 1 SD unit increase in metabolites.

## Data Availability

Data are available upon request and approval by the UK Biobank (https://www.ukbiobank.ac.uk/enable-your-research/apply-for-access (accessed on 20 July 2022)). Code book and analytic code will be made available upon request and approval by the corresponding author.

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
