# Peer review of "Exploration of Metabolic Biomarkers Linking Red Meat Consumption to Ischemic Heart Disease Mortality in the UK Biobank"

_nutrients, 2023, doi:10.3390/nu15081865_

Round 1

Reviewer 1 Report

Overview

The manuscript titled "Exploration of metabolic biomarkers linking red meat consumption to ischemic heart disease mortality in the UK Biobank" describes an in-depth examination of the relationship between red meat consumption and ischemic heart disease (IHD). This study is very interesting, with a clear purpose and a certain degree of innovation. However, the author requires to complete the following queries before publication.

Comments

1. It is well known that the total bioactive peptides of meat have been shown to affect cardiovascular function, but the metabolites mentioned in this study do not seem to be related, please elaborate on the reasons.

2. The metabolites screened in this study are all related to lipids. Please determine whether the research method is non-targeted metabolomics or lipid metabolomics?

3. Line 244 change “cardiovascular disease” to “cardiovascular diseases”

4. Line 267 change “amino acid” to “amino acids”

5. Line 292 “from the UK biobank”

6. Line 300 “at the repeat visit” please delete “at”

7. Line 303-312 This statement is much too far reaching. You cannot provide advice to the European population based on data from the study. This statement should be revised so it summarizes the study and doesn’t make unsupported over-reaching statements.

Author Response

Thank you very much for your helpful suggestions. Please see the attached file for the detailed point-to-point response.

Reviewer 2 Report

Dear Authors, 

Please find my observations below:

What type of red meat do you recommend based on this study? Because, based on this study, you concluded that red meat should be restricted in European countries. Why? European countries are the largest red meat consumers? 

Other observation is that your study is based on UK population. Why do you recommend to reduce the consumption of red meat in Europe? Did you checked some databases to follow the red meat consumption in UK versus Europe, and Europe versus USA, China and so on…

Although the general meat consumption is concerning somehow due to other reasons, the European Commission forecasts that EU meat consumption will decline slightly, from 69.8 kg to 68.7 kg per capita by 2030, because of growing social and ethical concerns, environmental and climate worries and health claims, but also because of the ageing population (eating smaller portions) and lower meat availability on the domestic market.

According to a post of Statista the biggest consumers of meat are Spain (99), Portugal (95), Iceland (91), Poland, (88) and Austria (87). Not only are Spain, Portugal and Iceland one of the biggest consumers of meat in Europe. They also are one of the biggest consumers of seafood in Europe. With a consumption of 92 kg of seafood per capita, Icelandic people consume more or less as much seafood as they consume meat! In 2019, the United Kingdom was declared to have about 61.5kg per capita.

So, in this light, do the authors consider that only red meat consumption cause IDH, CVD or CHD in UK population, when they are not top red meat consumers.

Do the authors consider that maybe other factors like, alcohol, smoking, not having a regular physic activity are also significant contributors to cause these affections among UK population?  

Please revise the references according to the instruction for authors.

Good luck!

Author Response

Thank you very much for the helpful suggestions. Please see the attachment for the point-to-point response.

Round 2

Reviewer 1 Report

thanks for your excellent responses.